# Characterization and Antimicrobial Activity of Silver Nanoparticles Synthesized with the Peel Extract of Mango

**DOI:** 10.3390/ma14195878

**Published:** 2021-10-08

**Authors:** Yage Xing, Xingmei Liao, Xiaocui Liu, Wenxiu Li, Ruihan Huang, Jing Tang, Qinglian Xu, Xuanlin Li, Jinze Yu

**Affiliations:** 1Key Laboratory of Grain and Oil Processing and Food Safety of Sichuan Province, College of Food and Bio-Engineering, Xihua University, Chengdu 610039, China; xingyg@mail.xhu.edu.cn (Y.X.); 212020095100038@stu.xhu.edu.cn (X.L.); 18408248463@163.com (W.L.); huangr_h@163.com (R.H.); TJ19980809@163.com (J.T.); xuqinglian01@163.com (Q.X.); xllb0519@163.com (X.L.); 2Key Laboratory of Food Non Thermal Technology, Engineering Technology Research Center of Food Non Thermal, Yibin Xihua University Research Institute, Yibin 644004, China; 3Department of Agricultural Technology, Neijiang Vocational and Technical College, Neijiang 641000, China; 4National Engineering Technology Research Center for Preservation of Agricultural Products, Key Laboratory of Storage of Agricultural Products, Ministry of Agriculture and Rural Affairs, Tianjin Key Laboratory of Postharvest Physiology and Storage of Agricultural Products, Tianjin 300384, China; 13032291162@126.com

**Keywords:** mango peel, silver nanoparticles, green synthesis, antibacterial activity

## Abstract

The green synthesis of silver nanoparticles (AgNPs) from biological waste, as well as their excellent antibacterial properties, is currently attracting significant research attention. This study synthesized AgNPs from different mango peel extract concentrations while investigating their characteristics and antibacterial properties. The results showed that the AgNPs were irregular with rod-like, spherical shapes and were detected in a range of 25 nm to 75 nm. The AgNPs displayed antibacterial activity against *Escherichia coli (E. coli**)* and *Staphylococcus aureus (S. aureus)*, showing a more significant impact when synthesized with 0.20 g/mL of mango peel extract. Therefore, the antibacterial effect of different diluted AgNP concentrations on the growth kinetic curves of *E. coli* and *S. aureus* after synthesis with 0.20 g/mL mango peel extract was analyzed. The results indicated that the AgNP antibacterial activity was higher against *S. aureus* than against *E. coli*, while the AgNP IC50 in these two strains was approximately 1.557 mg/mL and 2.335 mg/L, respectively. This research provides new insights regarding the use of postharvest mango byproducts and the potential for developing additional AgNP composite antibacterial materials for fruit and vegetable preservation.

## 1. Introduction

Due to improved living standards in recent years, fruit and vegetable sales have gradually increased, disseminating from a local scale to a national and even global scale, requiring enhanced storage conditions for fresh produce. Nevertheless, the annual incidence of postharvest fruit and vegetable decay in China is 20–40%, while the annual global loss ranges between 30–50% [1]. Fruit and vegetable degradation not only presents a threat to human health but also causes enormous economic losses in the food industry. *Escherichia coli (**E. coli)* and *Staphylococcus aureus (**S. aureus)* represent the primary foodborne pathogenic bacteria, and their persistence and distribution via fruits and vegetables have caused significant concern [2]. Pathogenic bacterial inhibition is essential for the postharvest preservation of fresh produce, highlighting the antibacterial advantages of nanomaterials in food preservation.

Nanomaterials have attracted substantial scientific interest due to their potential technological applications in food storage and preservation, catalysis, environmental protection, and bioassays, as well as in the medical, textile, and cosmetics industries [3,4,5,6,7,8,9,10]. Various metal- and metal oxide-based nanocomposites, including nano-Ag, nano-TiO_2_, nano-SiO_2_, and nano-ZnO are used as antimicrobial agents [11]. In the food industry, nanomaterials are generally incorporated into packaging material to improve its tensile characteristics and increase its antibacterial properties [12]. Silver nanoparticles (AgNPs) and nanocomposites have attracted attention for preserving the freshness of fruits and vegetables due to their large surface areas, strong stability, and broad-spectrum antibacterial properties. Therefore, it is vital to examine the preparation and antibacterial ability of AgNPs to enhance their application and development.

The main methods currently used for AgNPs synthesis include the physical, chemical, and biological methods [13,14,15,16]. Although simple to use, the physical method presents disadvantages, such as high cost and extensive energy consumption [17]. The chemical method can easily control reaction conditions but negatively affects human health and the environment [18]. Therefore, the focus has shifted to green methods for AgNP synthesis because they are simple, cost-efficient, and environmentally friendly [19,20,21]. One of these refers to the biological method, involving the use of innocuous, pollution-free, and harmless raw materials to prepare AgNPs [22]. Biological methods have the advantages of low energy consumption, low cost, and simple synthesis compared to physical methods. On the orher hand, biological methods have the advantages of green, renewable, and mild reaction conditions compared to chemical methods [23]. Green AgNP synthesis using plant extracts is considered environmentally friendly, cost-efficient, and fast while consuming little energy.

Plant extracts used as mediators for synthesizing AgNPs, include *Citrus limetta* peel, mangosteen leaf extract, *Ailanthus altissima* skin extract, *Sapindus emarginatus* pericarp, and *Ixora brachypoda* leaf extract [24,25,26,27,28]. Mangoes are known as the “King of Tropical Fruits” and belong to the Lacertidae family. They are widely cultivated in India, Bangladesh, the Central South Peninsula, Malaysia, and China. According to the FAO, global mango production in 2014 exceeded 45 M tonnes. Peel is the major by-product of industrial mango processing and makes up 7–24% of the total mango weight [29]. Although a significant amount of mango peel waste is generated worldwide, minimal studies are available regarding its high utilization value and the preparation of AgNPs using mango peel extract. Mango peels are rich in flavonoids, polyphenolic compounds, ascorbic acid, and other natural substances. Because it contains many hydroxyl groups, mango peels are natural reducing agents [30]. Green AgNP synthesis using mango peel extract is not only renewable but also environment-friendly.

Deriving nanoparticles from plant extracts can catalyze the decomposition of ethylene, successfully inhibiting bacterial growth. Several studies have investigated the preparation of composite antibacterial coating materials using nanomaterials. Lan et al. [31] prepared a composite antimicrobial film using sodium alginate, apple polyphenols, and AgNPs, extending the shelf life of strawberries by about 8 d at 4 °C and effectively maintaining their storage quality. An et al. [32] reported that AgNP-PVP coatings exhibited excellent preservation and microbial control effects on green asparagus. Similarly, Shankar et al. [33] indicated that chitosan/essential oil/AgNP composite membranes prepared using tyrosine as a reducing agent showed strong antibacterial activity against *E. coli*, *Listeria monocytogenes*, *Salmonella typhimurium*, and *Aspergillus niger*. Xing et al. [34] reviewed the preparation, antibacterial properties, and mechanisms of edible coatings containing antibacterial nanomaterials and their efficient applications to vegetables and fruits. However, AgNPs synthesized from different plants varied in shape and exhibited diverse antibacterial activity against Gram-positive and Gram-negative bacteria [35].

When exploring new AgNPs preparation methods, it is necessary to determine the antibacterial effect and the strength of antibacterial activity. Previous studies have shown that AgNPs with different sizes and morphologies present a diverse antibacterial effect [36]. Consequently, this study uses mango peel extract as a reducing agent for green AgNP synthesis while evaluating the antibacterial properties.

## 2. Materials and Methods

### 2.1. Materials

Fresh, ripe mangoes were obtained from the market in Panzhihua City, Sichuan Province, China. Silver nitrate (AgNO_3_) and sodium dodecyl sulfate were provided by the Chengdu Kelong Chemical Reagent Factory (Chengdu, China). Agar powder was provided by Zhejiang Hangzhou Microbial Reagent Co., Ltd. (Hangzhou, China). Beef extract, tryptone, and yeast extract were purchased from Beijing Aoboxing Biotechnology Co., Ltd. (Beijing, China). *S. aureus* and *E. coli* were provided by the School of Food and Bioengineering of Xihua University (China).

### 2.2. Green AgNP Synthesis

#### 2.2.1. Preparation of the Mango Peel Extract

The mango peels were collected and washed thoroughly using distilled water to remove the dust particles adhering to the surface and cut into small pieces. The mango peels (10 g, 15 g, 20 g, 25 g, 30 g, and 35 g) were transferred to 100 mL of distilled water and pressed to obtain an extract using a juicer. The extract was boiled at 85 °C for 30 min to obtain polyphenols substances from the mango peels [37]. These substances were used as reducing and capping agents to synthesize the AgNPs, which were then centrifuged at 2000 rpm for 10 min. The supernatant was filtered through 0.45-micron-pore-size membranes to obtain the extract, which was stored at 4 °C for further use.

#### 2.2.2. AgNP Synthesis

The AgNPs were prepared using AgNO_3_ as a precursor and mango peel extract as a reducing agent according to a method delineated by Turunc et al. [38], with slight modifications. A mixture consisting of 0.15 g sodium dodecyl sulfate, 25 mL 2.0 mmol/L AgNO_3_ solution, and 2 mL mango peel extract was prepared at concentrations of 0.10 g/mL, 0.15 g/mL, 0.20 g/mL, 0.25 g/mL, 0.30 g/mL, and 0.35 g/mL. The nanometer silver colloid was then obtained via heating and stirring for 1 h at 80 °C. A change in color from colorless to reddish-brown of the suspensoid certificated the green synthesis of AgNPs [39]. The reaction solution was centrifuged three times at 10,000 r/min for 10 min using a high-speed centrifuge. The subsequent precipitate was vacuum freeze-dried to obtain the AgNP powder, which was stored in anhydrous ethanol.

### 2.3. Transmission Electron Microscopy (TEM)

The morphology of the synthesized AgNPs was observed via TEM according to a method described by Li et al., with minor modifications [40]. A drop of the uniformly dispersed AgNP aqueous solution was deposited on a carbon-coated copper grid. The samples were dried completely before analysis, while the shapes and sizes of the AgNPs were analyzed using TEM at an acceleration voltage of 200 kV and a resolution of 0.24 nm.

### 2.4. Determining the Relationship between the Absorbance and Bacterial Concentrations

The bacterial suspensions were prepared using a technique outlined by Wang et al., with slight modifications [41]. The average number of colonies on three plates at the same dilution level was obtained by counting the dilution plates while calculating the bacterial suspension concentrations with different absorbance values. A correlation was generally evident between the concentrations and absorbance of the *E. coli* and *S. aureus* bacterial solutions [42,43]. The relationship curves between the absorbance and concentrations of the bacterial solutions were plotted with the absorbance as the abscissa and the bacterial colonies as the ordinate. An equation was used to calculate the bacterial concentrations as follows:$$c=\frac{n\times N}{V}$$

Here, *c* represents the bacterial concentration, *n* is the number of colonies, *N* denotes the dilution factor, and *V* signifies the sampling volume.

### 2.5. Antibacterial Activity against E. coli and S. aureus

The AgNP sols were prepared according to a method described by Vijayaraghavan et al. [44], with minor modifications. Each nanometer silver colloid was prepared using two different concentrations (40% and 20% of the original nanometer silver colloid solution) and the mango peel extract displaying the strongest antibacterial effect. The impact of different nanometer silver colloid concentrations on the growth curves of *E. coli* (5.0%, 10%, 20%, 30%, 40%, 50%, 60%, 70%, and 80% of the original nanometer silver colloid solution) and *S. aureus* (2.5%, 5.0%, 10%, 15%, 20%, 30%, 40%, 50%, and 60% of the original nanometer silver colloid solution) were determined to explore the half lethal concentration on these two bacterial strains.

The antibacterial effect of AgNPs on *E. coli* and *S. aureus* was evaluated using ultraviolet spectrophotometry with slight modifications [45]. Here, 5 mL AgNP sol and 3 mL of *E. coli* or *S. aureus* suspension at a concentration of 10^7^ CFU/mL were added to the sterile Luria-Bertani (LB) medium (LB, 5 g yeast extract, 10 g NaCl, 10 g tryptone, and 1000 mL water). The bacteria were cultured and maintained in LB broth and then incubated at 37 °C at 180 rpm, while the bacterial cell proliferation was monitored at intervals according to absorption at 600 nm [46]. Finally, the growth curve plot was created using the optical density (OD) value vs. time.

### 2.6. Scanning Electron Microscope (SEM) of E. coli and S. aureus

The morphological *E. coli* and *S. aureus* changes were observed via SEM using a method described by Gouveia et al., with slight modifications [47]. AgNPs were added to the *E. coli* and *S. aureus* (final concentration of IC50) suspensions, while a sterile buffer solution at pH 7.2 was added to the control group. The suspension was centrifuged at 10,000 r/min for 15 min, after which the supernatant was discarded. The bacteria were fixed with 2.5% glutaraldehyde for 2–4 h and washed three times with phosphate buffer. The aqueous ethanol solution was then subjected to gradient elution followed by centrifugation at 10,000 r/min for 5 min, after which the eluate was discarded. Before SEM observation, the sample was dried using a Leica EM CP300 automatic critical point dryer and sputter-coated with gold. An Apreo S-type scanning electron microscope was used to acquire the morphological, aggregate, and distribution information regarding the pre- and post-bacteriostatic changes.

### 2.7. Statistical Analysis

The experiments were carried out in triplicate, and data were expressed as means ± standard deviation. One-way analysis of variance (ANOVA) and Duncan’s multiple range tests were used to determine the significance of differences (*P* < 0.05) among means. The experimental data were acquired using origin software and analyzed using SPSS version 20.

## 3. Results and Discussion

### 3.1. TEM Results

The morphological structure is a vital indicator when evaluating AgNPs, and is directly related to the optical, electrical, and even biological properties. The morphology, dispersion, and sizes of AgNPs can be observed via TEM. As shown in Figure 1, different nanoparticle shapes were prepared by increasing the mango peel extract concentration while the synthesized AgNPs gradually increased. The AgNPs were generally spherical with a few larger ellipsoidal particles [48]. This variation in the sizes and shapes of biosynthesized nanoparticles is common. The AgNPs in this study were smaller than 100 nm, as shown in Figure 1. As illustrated in Figure 1a, the synthesized AgNPs were spherical and uniformly dispersed, varied in size, and displayed no agglomeration. Similarly, as can be seen from Figure 1b–e, the synthesized AgNPs were spherical, oval and had few irregular shapes. They were dispersed uniformly and there were also some aggregations. However, as shown in Figure 1f, AgNP shapes changed from spherical to rod-like, with distinct aggregation. The synthesized AgNP sizes ranged between 25 nm and 70 nm, which were smaller than those measured using a laser particle-size analyzer.

The results showed that different concentrations of mango peel extract yielded AgNPs with different morphological properties and particle sizes, which was consistent with the findings of previous studies. Ibrahim et al. [49] used banana peel extract (BPE) as a reducing and capping agent to prepare AgNPs, which were characterized using TEM. The TEM micrographs showed that the AgNPs were spherical and monodispersed, with an average size of 23.7 nm. Similarly, Gul et al. [50] derived AgNPs from *Ricinus communis* leaf and root extracts as reducing and stabilizing agents, while the TEM results confirmed that the subsequently synthesized particles were subspherical with mean sizes of 29 nm and 38 nm, respectively. Additionally, Padalia et al. [51] synthesized AgNPs by reducing an AgNO_3_ aqueous solution with *Tagetes erecta* flower broth as a reductant. TEM analysis indicated that the AgNPs were spherical, hexagonal, and irregular, with an average diameter of 10–90 nm. Contrarily, this study obtained an average AgNP size of less than 100 nm, indicating that the mango peel extract reduced the AgNO_3_ to successfully prepare AgNPs. The AgNP size distribution was 25–75 nm, showing that different plants exhibited different reducibility, while the parameter conditions, such as extract concentration, temperature, and contact time, were inconsistent [52]. Therefore, AgNPs with a small particle size could be obtained using mango peel extract as a raw material. The varied morphology and particle sizes of the synthetic AgNPs could be attributed to the diverse aggregation effect of different mango peel extract concentrations. Because the mango peel was rich in polyphenols, it acted as a capping agent, reducing the AgNO_3_ to coat the AgNP surface. Therefore, the aggregation was ascribed to the layer (capping agents) covering the AgNPs, causing them to adhere to each other [53]. The particle sizes of the AgNPs increased in conjunction with higher mango peel extract concentrations and shape changes, which was related to the crystal nucleus growth rate [54]. A lower mango peel extract concentration decelerated the AgNP nucleation rate, allowing more significant nucleation to occur simultaneously. The final product tended to form sphere-like bodies when the growth rate difference between the individual crystal planes in the system was low. An increased mango peel extract concentration extended the AgNP synthesis time. AgNP proliferation was promoted by a slow reaction time, yielding a small number of large, individual nanoparticles. Further increasing the mango peel extract concentration accelerated the overall reaction rate of the system, causing the nuclei to grow in a specific direction and allowing the formation of the rod-shaped AgNPs [55].

### 3.2. The Effect of AgNPs Synthesized Using Different Mango Peel Extract Concentrations on E. coli and S. aureus

The bacterial proliferation rate in the culture medium was reflected by the absorbance value [56]. *E. coli* and *S. aureus* were used during the experiment as indicator bacteria to determine the inhibitory effect of the different AgNPs obtained via mango peel extract on their growth. The effect of different AgNPs on the antibacterial activity of *E. coli* and *S. aureus* are shown in Figure 2 and Figure 3, respectively. Although a significant (*p* < 0.01) bactericidal effect was evident, differences were apparent between the two strains. The antibacterial activity first increased, followed by a decline at a higher extract concentration. Moreover, as shown in Figure 2a and Figure 3a, rapid *E. coli* proliferation was observed within 6 h of adding a low-concentration AgNP solution, while *S. aureus* reached maximum growth within 8 h. Therefore, increasing the AgNP concentration could prolong the delay period of microbial growth. The antibacterial activity first increased, followed by a decrease in conjunction with a higher extract concentration. The bactericidal effect of the synthesized AgNPs against *E. coli* and *S. aureus* was significant (*p* < 0.01) at a mango peel extract concentration of 0.20 g/mL, reaching its maximum performance against these two strains.

The results showed that increasing the mango peel extract concentration did not necessarily improve the antibacterial effect of the prepared AgNPs. Instead, optimal AgNP bactericidal performance relied on the cooperation between particle size, morphology, and the number of particles [57]. However, these properties may be affected by the mango peel extract concentration, in turn impacting the antibacterial effect of the AgNPs. Mango peel extract contains many reducing substances, such as flavonoid compounds, polyphenolic compounds, and ascorbic acid. These substances have different functional groups, such as hydroxyl and aldehyde groups, which can be used as capping agents to facilitate reduction and stability and change the shapes and sizes of the AgNPs [30]. Patil et al. [58] indicated that biomacromolecules in plant material could increase AgNPs stability while effectively preventing precipitation formation via aggregation. However, the silver ion (Ag^+^) reduction rate increased in conjunction with a higher mango peel extract concentration, resulting in AgNP aggregation and precipitation [59]. At lower mango peel extract concentrations (0.10 g/mL and 0.15 g/mL), fewer, small-sized synthesized AgNPs were evident with lower antibacterial properties. The particle sizes and the number of synthesized AgNPs increased in conjunction with higher mango peel extract concentrations, contributing to their increased antimicrobial properties. Nevertheless, with a further increase in the mango peel extract concentration (0.25 g/mL, 0.30 g/mL, and 0.35 g/mL), the Ag^+^ reduction rate became too high, rapidly increasing AgNP aggregation, possibly reducing the antibacterial activity. This was in line with Song [14], who showed there was an increase in the synthesis rate of AgNPs with the increase in *Magnolia kobus* leaf extract. The accumulation of more reduction groups on the nuclear surface reinforced the secondary Ag^+^ reduction. Therefore, a mango peel extract concentration of 0.20 g/mL represented the preferred conditions for antibacterial AgNP preparation.

### 3.3. The Effect of Different AgNP Concentrations on E. coli and S. aureus

The antibacterial effect of different AgNP sol on *E. coli* and *S. aureus* were analyzed, considering 0.20 g/mL as the optimal AgNP sol. As shown in Figure 4, the experimental group exhibited variation regarding the antibacterial effect against *E. coli* and *S. aureus* at different AgNP concentrations compared with the OD value of the control group. As illustrated in Figure 4a, the *E. coli* growth curve was close to that of the control at a 5.0% stock solution and AgNP concentration, indicating weak antibacterial activity against this strain. However, the *E. coli* proliferation tended to be horizontal at an 80% stock solution concentration, indicating that this AgNP concentration completely inhibited *E. coli* growth. Consequently, the minimum inhibitory concentration (MIC) required for *E. coli* restriction was about 80% AgNP precursor sol, while the fastest *E. coli* growth rate was observed after 6 h of incubation. As shown in Figure 4a, the survival rate of *E. coli* was 50% when the concentration of nano-silver was 30% of the concentration of nano-silver solution, which meant that the IC50 of nano-silver on *E. coli* was about 30% of the concentration of nano-silver solution. Similarly, the fastest *S. aureus* proliferation was evident within 8 h, except at 40%, 50%, and 60% concentrations, as shown in Figure 4b. An AgNP solution of about 60% represented the MIC against *S. aureus*, while a 20% AgNP solution represented the half lethal concentration. The results indicated that *E. coli* was restricted by an AgNP MIC of about 8.303 mg/L and a median lethal concentration of approximately 3.113 mg/L. In addition, *S. aureus* was inhibited by an AgNP sol MIC of about 6.227 mg/L and a median lethal concentration of approximately 2.067 mg/L.

Although the synthesized AgNP solution displayed prominent antibacterial activity against *E. coli* and *S. aureus*, their effects differed because *S. aureus* was probably more sensitive to AgNPs than *E. coli* [60]. Furthermore, using MIC tests and standard growth curves, we have shown that increasing concentrations of nano-silver colloid can reduce the growth of bacteria. The bactericidal activity against *S. aureus* was generally slightly higher than *E. coli*, which was consistent with the results of previous studies. Balazi et al. [61] synthesized AgNPs using *Lavandula angustofolia* L. as a reducing agent and evaluated their antibacterial activities against *S. aureus* and *E. coli*, showing a slightly higher inhibitory effect against *S. aureus* than *E. coli*. Similarly, Sharifi-Rad et al. [62] synthesized AgNPs using *Otostegia persica* (Burm.) Boiss leaf extract as a reducing agent and obtained the same results. Furthermore, Rafique et al. [63] derived AgNPs from *Albizia procera* leaf extract. The results showed that AgNPs were significantly effective in inhibiting *S. aureus* and *E. coli,* exhibiting higher activity against *S. aureus*. Paul et al. [64] reported that metal nanoparticles destroyed the exterior cell wall and exudate cell materials so that *S. aureus* might leak more proteins. It seemed possible that the variety of cell walls between *S. aureus* and *E. coli* was a major cause. A vital function of the outer membrane is acting as a protective barrier, preventing or slowing the entry of antimicrobial agents and other toxic substances that may kill or harm the bacteria. The *E. coli* cell wall consisted of an outer membrane composed of 50% lipopolysaccharides, 35% phospholipids, and 15% lipoprotein and was about 6–18 nm thick. This outer film acted as a carrier and protected against the antimicrobial agent [65]. On the contrary, it might be due to the fact that *S. aureus* lacked a protective membrane around the peptidoglycanlayers in the cell wall. This allowed easier interaction between the nanoparticles and outer membrane until degradation occurred, inhibiting *S. aureus* proliferation more effectively [66].

### 3.4. SEM Observation of the E. coli and S. aureus Morphology Combined with AgNP Sol

Figure 5 shows representative images of *E. coli* and *S. aureus* before and after AgNP treatment at 20,000× magnification. As shown in Figure 5a, the *E. coli* surface in the negative control group was smooth, with intact, complete cells. However, the *E. coli* treated with AgNPs became rough and wrinkled, while the cell surface was damaged, as shown in Figure 5b. The *E. coli* cell membrane contracted, and the cell morphology was distorted. Figure 5c shows that the *S. aureus* surface in the negative control group was smooth and cellularly intact. As illustrated in Figure 5d, the surfaces of some *S. aureus* cells in the experimental group were cracked, indicating that the AgNPs adhered to the bacterial cell membrane, causing damage. Moreover, other granular substances were observed around *S. aureus* in the experimental group, speculated to be AgNPs [66]. Furthermore, Figure 5b,d show that the AgNP sols synthesized using mango peel extracts caused morphological changes in *E. coli* and *S. aureus*.

The SEM results indicated that the AgNPs had an obvious destructive effect on *E. coli* and *S. aureus*, causing changes in the surface structure and even bacterial disintegration. These results were consistent with previous studies. Xue et al. [67] prepared chitosan/AgNPs (CS/AgNPs) sols via the in situ synthesis of reduced AgNO_3_ within chitosan using punicalagin as a green reducing agent, disrupting the morphologies of *E. coli* and *S. aureus*. Similarly, Jiang et al. [68] biosynthesized AgNPs using *Piper sarmentosum* Roxb. extract. SEM characterization showed that the AgNPs caused morphological *E. coli* and *S. aureus* changes. Overall, SEM images of *E. coli* and *S. aureus* showed that AgNPs destroyed the surface structures of the bacteria. It is speculated that the antibacterial mechanism mainly includes the following three aspects: First, AgNPs could destroy cell membranes [35]. The synthesized AgNPs were all smaller than 150 nm, allowing easy absorption on the surface of the bacterial cell wall to react with the -NH-, and -COOH functional groups of the peptidoglycan, corroding the cell wall. However, the cell membranes were destroyed due to the antioxidant activity of the polyphenols in the mango peel extract, resulting in biomolecule leakage into the bacterial cells [60]. In addition, cellular respiration may be inhibited by AgNPs. The AgNPs disrupted the cell wall, penetrating the cell and rapidly binding to the thiol groups of the intracellular enzyme proteins, inactivating the enzymes and leading to cell metabolism failure and death [69]. Finally, the AgNPs damaged the cellular DNA and ribosome assembly [70]. Furthermore, the AgNPs may inhibit the growth of microorganisms by acting on cell DNA. The hydrogen bonds in the adjacent bacterial purines replaced the silver in the AgNPs, while DNA could not repair itself, resulting in protein inactivation and death to achieve sterilization.

## 4. Conclusions

This study presents an eco-friendly, cost-effective method to synthesize AgNPs from mango peel extract. AgNPs varying in particle size and shape are successfully prepared using different mango peel extract concentrations as reducing agents, demonstrating significant antibacterial activity against *E. coli* and *S. aureus*. This bactericidal action is more successful against *S. aureus* than against *E. coli*. AgNPs can destroy cell walls, restrict enzyme respiration, and damage the DNA replication path to achieve sterilization. However, in order to obtain AgNPs with an ideal particle size, morphology, and good antibacterial properties, it is important to compare them with AgNPs prepared by traditional methods. Moreover, the antibacterial effect of synthesized AgNPs is also affected by food components, media, and other factors. Therefore, further research is necessary for the adequate investigation of these issues. In conclusion, the AgNPs synthesized using mango peel as a reducing agent show significant potential for developing composite antibacterial AgNP materials for fruit and vegetable preservation.

## Figures and Tables

**Figure 1 materials-14-05878-f001:**
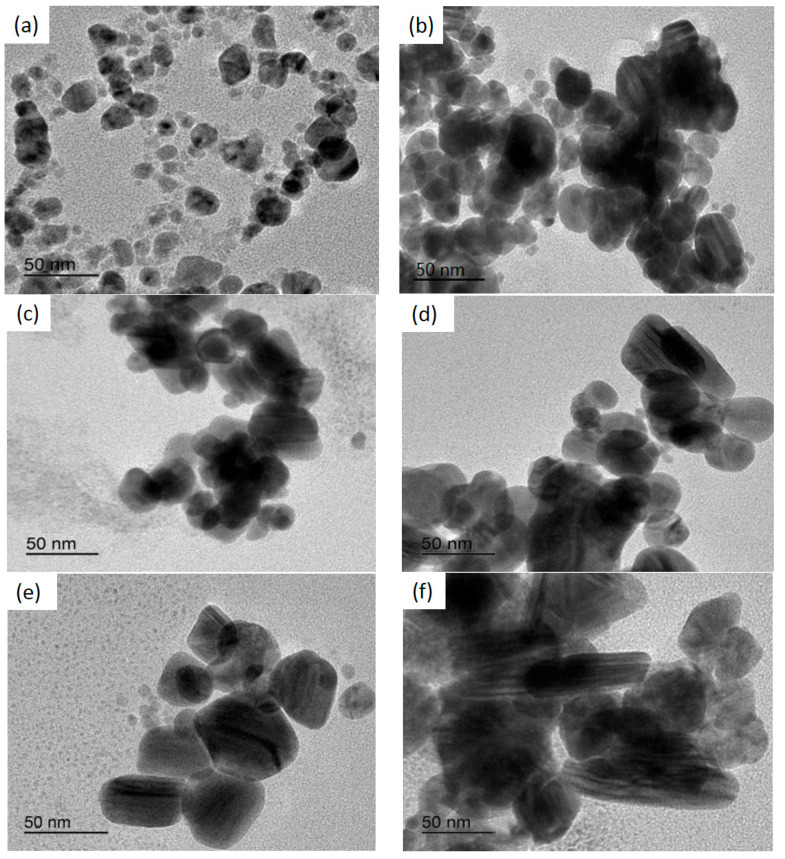
TEM observation of AgNPs synthesised using different concentrations of mango peel extract ((**a**) 0.10 g/mL mango peel extract, (**b**) 0.15 g/mL mango peel extract, (**c**) 0.20 g/mL mango peel extract, (**d**) 0.25 g/mL mango peel extract, (**e**) 0.30 g/mL mango peel extract, (**f**) 0.35 g/mL mango peel extract).

**Figure 2 materials-14-05878-f002:**
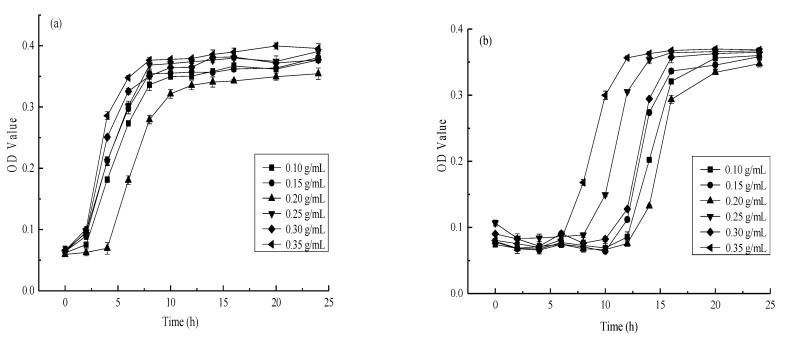
Effects of different silver nanoparticles on the antibacterial activity of *E. coli* ((**a**) Nanosilver sol concentration is 20% nanosilver sol stock solution, (**b**) Nanosilver sol concentration is 40% nanosilver sol stock solution).

**Figure 3 materials-14-05878-f003:**
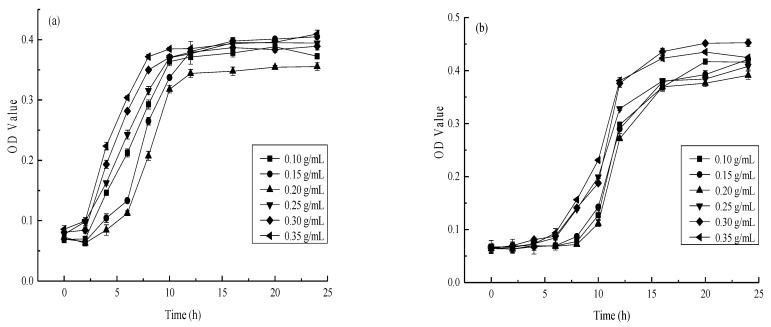
Effects of different silver nanoparticles on the antibacterial activity of *S. aureus* ((**a**) Nanosilver sol concentration is 20% nanosilver sol stock solution, (**b**) Nanosilver sol concentration is 40% nanosilver sol stock solution).

**Figure 4 materials-14-05878-f004:**
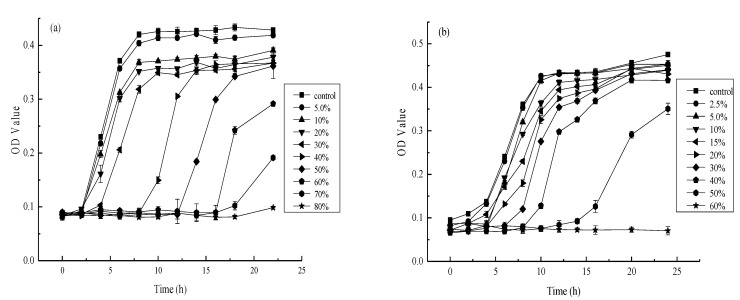
Antibacterial activity of microorganism under different concentrations of nano-silver colloid ((**a**) *E. coli*, (**b**) *S. aureus*).

**Figure 5 materials-14-05878-f005:**
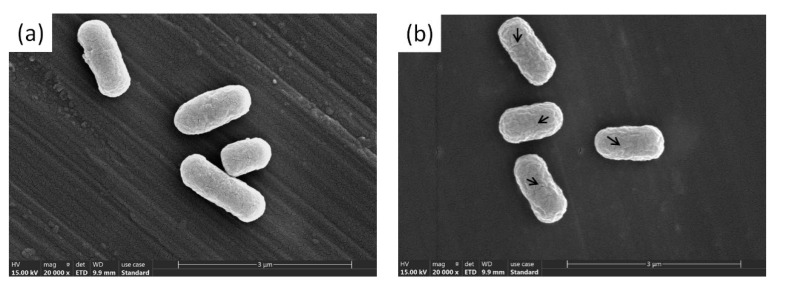
SEM observation of morphological changes of the tested bacteria ((**a**) *E. coli* treated with negative control magnified 20,000×, (**b**) AgNP-treated *E. coli* magnified 20,000×, (**c**) *S. aureus* treated with negative control magnified 20,000×, (**d**) AgNP-treated *S. aureus* magnified 20,000×).

## Data Availability

Not applicable.

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
