# Peer review of "Characterization and Antimicrobial Activity of Silver Nanoparticles Synthesized with the Peel Extract of Mango"

_materials, 2021, doi:10.3390/ma14195878_

Round 1

Reviewer 1 Report

1# In Introduction part, some papers should be cited to support the statement of “Currently, the main methods that are used for synthesising AgNPs include the physical method, the chemical method, and the biological method.” such as The Journal of Physical Chemistry 97.21 (1993): 5457-5471, https://doi.org/10.1039/C3NR05479A, and https://doi.org/10.1007/s00449-008-0224-6.

2# In 2.2.1, some details of experimental should be given. For example, how to press mango peel into juice? and why is “The mixture was boiled at 85 °C for 30 min”?

3# In 2.2.2, are the mango peel extracts enough to reduce silver nitrate?

4# In Figure 1, using mango peel extract, AgNPs can be synthesized. However, their morphologies are not uniform. Can the authors give some advices about how to obtain uniform nanoparticles using mango peel extract?

5# In Figure 5, the scale bars should be clear. Furthermore, if possible, please provide some high-resolution images. It is difficult to say the materials in Figure 5a have smooth surface, while the materials in Figure 5b have rough surface.

Reviewer 2 Report

This manuscript has studied the effect of mango peel extract on the biological synthesis of AgNP, and analyze the effect of mango peel concentration on AgNP size, aggregation, concentration, morphology, which could further affect the antibacterial effect on E.coil and S.aureus. The experiments were properly designed and the results were shown in good logic. However, in the result part, the author did not clearly make a reasonable discussion on the results. Although the author has stated that the result is reasonable because previous studies have shown similar results and conclusions, it will be more important to have your own discussion on the result by providing a reasonable explanation of the phenomenon and detailed analysis of the SEM/TEM images. I hope the author can improve the discussion on emphasizing the originality of this manuscript. In addition, the author might need to proofread the manuscript for a more straightforward expression and statement.

Line 49: please add reference for “nanomaterials can improve their tensile 49 properties”.

Line 207: Figure 1: (1) Is there FFT of TEM picture to confirm they are Ag NP? (2) According to the result from analyzer (or image processing, if possible), what is the particle size distribution? (3)  Is there a result showing how concentration can affect the particle size and circularity?

Line 216-226: The result is consistent with previous studies, and the author does not need to summarize the conclusion from other studies. Comparing with previous studies, what is the main reason that Ag NPs have an average size that is smaller than 100nm? And, what is the main reason that the result in this study shows 25-70nm in NP size?

Line 240: The author has analyzed the NP synthesis process, and clearly explain the size and shape. However, the author does not clearly clarify the cause of the aggregation phenomenon. Please elaborate.

Line 259: Figure 2 and 3, please add Optical Density (OD) when the author first mentions this acronym in the manuscript.

Line 280: The statement here is not clear. Can the author specify what concentration are defined as low and high?

Line 285: I can understand the statement that “high level of aggregation results in surface area reduction and therefore can decrease the antibacterial effect”, but why is high concentration resulting in decreasing of antibacterial effect?

Section 3.2 does not clearly state the relationship between “antibacterial effect” and “mango peel concentration”. I understand that mango peel concentration can directly affect a) AgNP size, b) NP concentration and c) particle shape/morphology. According to the study, what the major factor to antibacterial strength? And how comes 0.20g/mL the preferred condition?

Line 300-302: This statement is not supported by any of the information from Figure 4. As what the author describes in the previous sentence, Figure 4 shows the difference between experimental groups of different AgNP concentration, but not between AgNP synthesized with/without mango peel.

Line 363: the difference between figure a and b are not significant. This could be the result from imaging. Cell surface damaging cannot be clearly observed. Author could specifically indicate the damaged location or use additional method to show the damaged cells.

The author needs to proof-read the manuscript for proper use of language, a few suggestion:

  • Line 59: “…exhibits negative effects on both human health as well asand the environment”.
  • Line 65: “…as it retains low cost and consumes little energy, while being a much and rapid synthesis method.
  • Line 77: ”kind of environment” => “environment-friendly”
  • Line 202: “roughly spherical” => “oval”
  • Line 267: The author might consider rewriting the statement to “showed that higher concentration of mango peel extract does not necessarily provide a better antibacterial effect”.
  • And more.

Reviewer 3 Report

Yu and co-workers developed a synthesis of silver nanoparticles using the peels extract of mango fruits. They studied the antibacterial activity of the so-prepared NPs vs E. Coli and S. Aureus, two commonly studied bacteria. The idea of the manuscript is interesting, and the investigation of the antibacterial activities in function different parameters (such as amounts of mango peels used in the synthesis) is captivating. However, the manuscript is written in a poor English, and must be substantially corrected. I recommend contacting an English editing Service. Also, some important issues make the manuscript unacceptable for publication in the present form. I’d suggest a reconsideration for publication after addressing these points:

  • Please check some typos (e.g. E.coli sometimes is written as E.coil)
  • Line 55-57 please add some details: explain the differences between physical, chemical and biological methods.
  • Line 60, 61 more recent references are needed. (e.g. 10.3390/photochem1020009)
  • Please quantify the amount of peel mango produced as waste (line 72-72)
  • Line 103,104: the intro section claimed that the NPs can be produced using waste peels. However, for the synthesis the authors used fresh mango. I suppose that the polyphenols content in waste peels (after a few days, or at least hours) before utilization for the production of silver nanoparticles may change. Will the preparation of the particles be affected is waste peels are used?
  • Why did you boil the polyphenols at 85C? No literature is reported. You might cite works where 85C are selected as extraction T.
  • There is an important limit in the experimental section: the authors should do additional experiments for the determination of the antimicrobial activity of the NPs prepared via traditional methods and employed in the same reaction conditions reported (at least in the most relevant experiments).
  • In the conclusion section the synthesis is claimed to be eco-friendly and cost-effective. However, these characteristics were are not fully discussed in the main text
  • The authors must discuss the novelty of the manuscript in comparison with similar papers reported in the literature
  • The authors should add some more recent literature (research articles) related to silver nanoparticles and their use also in catalysis, to better highlight their potentialities in industry and to possibly expand the utilization of the particles prepared by the authors. For example: 10.3390/catal10111343, 10.1021/acssuschemeng.9b00198

Round 2

Reviewer 2 Report

The responses from the authors have clearly answered my questions and the manuscript is significantly improved in the aspect of "results and discussion" and proper language use. 

I suggest the publication of this manuscript.

Reviewer 3 Report

The authors did a good job and fully revised the manuscript. Although one issue was not addressed, the authors declare that they will work on it in the upcoming research works. All the other questions were answered. The quality of the manuscript has sensibly increased and the work deserves now to be published. I have only one last suggestion: I think that the title can be reformulated making it more captivating (may be a little shorter) and more adequate to the nice job presented by the authors. Please also increase a little the thickness of the circles in figure 5b, d and add at least a reference in line 66.
